# The miR-205-5p/BRCA1/RAD17 Axis Promotes Genomic Instability in Head and Neck Squamous Cell Carcinomas

**DOI:** 10.3390/cancers11091347

**Published:** 2019-09-11

**Authors:** Fabio Valenti, Andrea Sacconi, Federica Ganci, Giuseppe Grasso, Sabrina Strano, Giovanni Blandino, Silvia Di Agostino

**Affiliations:** 1Oncogenomic and Epigenetic Unit, Department of Diagnostic Research and Technological Innovation, IRCCS Regina Elena National Cancer Institute, 00144 Rome, Italy; fabio.valenti@ifo.gov.it (F.V.); andrea.sacconi@ifo.gov.it (A.S.); federica.ganci@ifo.gov.it (F.G.); misteri@iol.it (G.G.); 2Molecular Chemoprevention Group, Department of Diagnostic Research and Technological Innovation, IRCCS Regina Elena National Cancer Institute, 00144 Rome, Italy; sabrina.strano@ifo.gov.it

**Keywords:** miRNA, HNSCC, DNA repair, genomic instability, mutant p53

## Abstract

Defective DNA damage response (DDR) is frequently associated with tumorigenesis. Abrogation of DDR leads to genomic instability, which is one of the most common characteristics of human cancers. *TP53* mutations with gain-of-function activity are associated with tumors under high replicative stress, high genomic instability, and reduced patient survival. The *BRCA1* and *RAD17* genes encode two pivotal DNA repair proteins required for proper cell-cycle regulation and maintenance of genomic stability. We initially evaluated whether miR-205-5p, a microRNA (miRNA) highly expressed in head and neck squamous cell carcinoma (HNSCC), targeted *BRCA1* and *RAD17* expression. We found that, in vitro and in vivo, *BRCA1* and *RAD17* are targets of miR-205-5p in HNSCC, leading to inefficient DNA repair and increased chromosomal instability. Conversely, miR-205-5p downregulation increased *BRCA1* and *RAD17* messenger RNA (mRNA) levels, leading to a reduction in in vivo tumor growth. Interestingly, miR-205-5p expression was significantly anti-correlated with *BRCA1* and *RAD17* targets. Furthermore, we documented that miR-205-5p expression was higher in tumoral and peritumoral HNSCC tissues than non-tumoral tissues in patients exhibiting reduced local recurrence-free survival. Collectively, these findings unveil miR-205-5p’s notable role in determining genomic instability in HNSCC through its selective targeting of *BRCA1* and *RAD17* gene expression. High miR-205-5p levels in the peritumoral tissues might be relevant for the early detection of minimal residual disease and pre-cancer molecular alterations involved in tumor development.

## 1. Introduction

The exogenous and endogenous damage of DNA leads to the destruction of the cell’s genomic build. In these cases, eukaryotic cells activate a complex DNA damage response (DDR) that allows them to correctly activate cell-cycle checkpoints and implement DNA repair mechanisms [1]. In neurodegenerative disorders, immune deficiencies, aging, and cancer, these mechanisms fail, and genomic instability occurs [1,2]. Chromosomal instability (CIN) is common in many types of cancer and can be caused by replication stress, non-conservative mitosis, telomere instability, and defective repair of double-stranded DNA breaks (DSBs) [1,2]. DSB repair can be done via non-homologous end-joining (NHEJ) or homologous recombination (HR). DSB is involved in tumor development as it facilitates the loss of tumor suppressors and the activation of oncogenic pathways. HR is activated only during the synthesis (S) phase and the first gap 2 (G2) phase to repair the DSBs as a template DNA is needed. In human tumors, DNA damage repair is a priority at the expense of fidelity to ensure cell survival. It requires numerous factors, including the RAD17 and RAD51 recombinases and the breast/ovarian cancer susceptibility gene products, *BRCA1* and *BRCA2* [1,2,3].

The tumor suppressor p53 is a very important transcription factor that regulates the response to DNA damage, leading to the transcription of genes involved in cell-cycle arrest, DNA repair, or apoptosis [4,5]. In over half of all human cancers, the *TP53* gene presents missense mutations that destroy the tumor suppressive competency of the protein and confer new oncogenic capacities (gain-of-function (GOF) mutations) that give a growth advantage to the tumor cell [4,5,6]. One essential role of mutant p53 protein (mutp53) is to promote cellular transcription, acting as a co-factor that mediates specific transcription factors to the promoters of diverse genes [5]. Head and neck squamous cell carcinomas (HNSCCs) are characterized by high genomic instability and represent the sixth leading cancer worldwide with 600,000 cases annually reported [7]. HNSCCs have about 60% of mutations in the *TP53* gene, where a number of studies reported that *TP53* mutations are generally associated with shorter recurrence-free and/or overall survival in these cancers (http://www-p53.iarc.fr). DNA interstrand crosslink-induced chromosomal breakage or defects in sister chromatid cohesion have a high incidence in HNSCC [2,8]. In a few cases, defects in the Fanconi anemia pathway are responsible for mutations in genes related to chromatid cohesion genes [8].

MicroRNAs (miRNAs) are small non-coding RNAs, and 22 nucleotides in length, which are able to modulate gene expression at a post-transcriptional level. Recent studies identified them as specific markers for diagnosis, staging, and treatment of cancer [9]. MiRNAs function as both effectors and modulators of the DNA damage response, interconnected with bona fide DNA damage players, such as ATM, p53, and MMR [10]. Deregulated miRNAs can modulate many aspects of carcinogenesis, such as cell proliferation, cell-cycle control, apoptosis, metastatization, and angiogenesis [11]. Increasing evidence showed that cells lacking miRNA biogenesis exhibit abnormal cell-cycle checkpoints and DNA repair [12]. For example, miR-210 and miR-373 target *RAD52* and *RAD23B* expression, respectively, and contribute to regulating nucleotide excision repair and HR repair in hypoxic conditions [13]. MiR-24 downregulates histone H2AX expression, thereby modulating cellular response to multiple DNA damaging events [14].

We recently assessed that mutant p53 also regulates the expression of miRNAs, such as miR-128b-5p and miR-223 [15,16]. Furthermore, we identified 49 miRNAs whose expression distinguishes tumors with wild-type p53 from those with mutated p53 protein in an HNSCC patient cohort [17]. Our previous findings showed that mutp53 sustains high expression levels of miR-205-5p in HNSCC [18] and that miR-205-5p belongs to a signature of 12 miRNAs regulated by mutp53 that predicts the risk of recurrence in HNSCCs [19].

In this study, we aimed to dissect the mechanistic role of the oncogenic activities of miR-205-5p in the induction and maintenance of genomic instability in HNSCC. Importantly, we documented that miR-205-5p was an important player in the mechanism of DDR. We reported that miR-205-5p targeted the expression of *BRCA1* and *RAD17*, which led to the inhibition of DNA repair activity and to the concomitant increase of replicative stress DNA damage. Consistently, we found a significant reduction in tumor growth and increased levels of *BRCA1* and *RAD17* in xenografted tumors arising from the HNSCC CAL27 cell line, orthotopically and intrascapularly injected in immunocompromised mice and subsequently treated with miR-205-5p synthetic inhibitors. Furthermore, we showed that high levels of miR-205-5p anti-correlated with *BRCA1* and *RAD17* expression in HNSCC patients. Expression of miR-205-5p was significantly higher in HNSCC peritumoral samples compared to matched non-tumoral tissues.

Collectively, our findings highlight the in vitro and in vivo oncogenic role of miR-205-5p in HNSCC.

## 2. Results

### 2.1. Mir-205-5p Targets the BRCA1 and RAD17 DNA Repair Genes

We previously reported that *BRCA1* and *RAD17* genes were transcriptionally repressed by the oncogenic mutant p53/E2F4 protein complex which was recruited onto the E2F4 consensus sequences within the *BRCA1* and *RAD17* promoter regions [20]. This led to an inefficient DNA repair activity and caused DNA damage accumulation during cell proliferation [20]. To shed light on additional mechanisms via which DNA repair activity was inhibited in HNSCC, we focused our attention on miR-205-5p. In HNSCC patients, we observed that miR-205-5p belongs to a signature of 12 miRNAs strongly associated with recurrence, turning out to be the best predictor of survival [19].

Reassured by these arguments, we adopted a computational approach by using the available prediction algorithm MiRWALK 3.0 to identify protein targets of miR-205-5p [21,22]. Interestingly, among miR-205-5p putative targets, we found that gene products such as *BRCA1*, *RAD17*, *NF-κB*, and *E2F5* were already reported as mutant p53 transcriptional targets [15,20,23]. We evaluated miR-205-5p as a non-coding molecule capable of inhibiting the expression of the DNA repair *BRCA1* and *RAD17* factors (Appendix A) thereby amplifying the previously documented mechanism of mutant p53 transcriptional repression of *BRCA1* and *RAD17* gene promoters (Figure 1A) [20].

We transfected mimic miR-205-5p in p53-null A253 HNSCC cells as they have low levels of miR-205 (Figure 1B) and observed the downregulation of BRCA1 and RAD17 proteins (Figure 1C), as well as for the other selected previously validated targets E2F5 and NF-κB proteins (Appendix A). We also found that miR-205-5p mimic transfection did not affect *BRCA1* and *RAD17* transcript levels (Appendix A).

Accordingly, an in silico analysis revealed that a seed-targeting sequence for miR-205-5p resides in the 3’ UTR (untraslated region) of both *BRCA1* and *RAD17* mRNAs, inducing a direct regulation of miR-205-5p on their expression (Figure 1D). To validate this, the DNA fragments of *BRCA1* and *RAD17* 3’ UTRs, enclosing the binding site for miR-205-5p (seed sequence), were cloned in a luciferase reporter vector downstream of the luciferase sequence (Appendix A). These constructs were transfected together with mimic miR-205-5p or a mimic control in the A253 cell line. The luciferase reporter assay showed that both *BRCA1* and *RAD17* 3’ UTRs were specifically targeted by mimic miR-205-5p (Figure 1E). As a negative control, we transfected the DNA fragments of *BRCA1* and *RAD17* 3’ UTRs where the binding site for miR-205-5p (seed sequence) was mutated (Figure 1E).

Subsequently, we examined whether the depletion of endogenous miR-205-5p affected the expression of BRCA1 and RAD17 proteins. To this end, CAL27 head and neck cancer cells (endogenously expressing mutp53A193T) were transfected with an LNA (locked nucleic acid)-modified 8-mer seed-targeting anti-miR-205-5p in order to deplete miR-205-5p (Figure 1F). We observed that the downregulation of endogenous miR-205-5p caused an increase in luciferase activity of *BRCA1* and *RAD17* 3’ UTR luciferase (Luc) vectors (Figure 1G), while the depletion of miR-205-5p did not modulate the luciferase activity of seed-sequence mutated *BRCA1* and *RAD17* 3’ UTR Luc-vectors (Figure 1G). This result paired with the increase in endogenous BRCA1 and RAD17 protein expression (Figure 1H) without affecting their mRNA expression (Appendix A, Appendix A). Additionally, we evaluated whether miR-205-5p changed the strength of Ago2 interaction, a protein required for RNA-mediated gene silencing (RNAi) by the RNA-induced silencing complex (RISC), with BRCA1 and RAD17 transcripts. The RNA immunoprecipitation assay (RIP) in A253 cells overexpressing miR-205-5p showed a strong interaction of Ago2 with both RAD17 and BRCA1 transcripts, supporting the notion that high miR-205-5p expression levels decreased *RAD17* and *BRCA1* expression (Figure 1I).

Altogether, these results highlighted that miR-205-5p directly targets *BRCA1* and *RAD17* expression in HNSCC cell lines.

### 2.2. Inhibition of Mir-205-5p Impairs the Tumoral Features of HNSCC Cell Lines

We next investigated the potential involvement of miR-205-5p in the aberrant cell proliferation, colony formation ability, and cell motility of HNSCC cells. To this end, we monitored the amount of cellular ATP until 72 h as an index correlated with metabolic activity and proliferation. At 48 h after transfection, CAL27 depleted of miR-205-5p proliferated much less than the control cells (Figure 2A). This result paired with a decrease in gap 1 (G1)-phase and S–G2-phase cell accumulation as revealed by the cytofluorimetric analysis (Figure 2B; Appendix A). The BrdU incorporation assay showed that LNA–miR-205-5p transfection caused a lengthening of the S phase (Figure 2C). We also observed that miR-205-5p depletion strongly affected the formation of colonies of CAL27 cells (Figure 2D). Furthermore, the in vitro transwell invasion assay in CAL27 showed that depletion of miR-205-5p expression decreased cell migration (Figure 2E).

Three-dimensional (3D) spheroids represent an ascertained system to study the tumor environment, as they allow cell–cell and cell–matrix contact. Here, we aimed to assess the impact of miR-205-5p depletion on CAL27 3D spheroids [24]. To this end, CAL27 cells were seeded in the alginate hydrogel matrix and transfected with LNA miR-205-5-p and LNA control oligos to form 3D spheroids (Figure 2F,G). After 10 days, we observed that miR-205-5p downregulation (Figure 2F) severely affected the ability of CAL27 cells to form well-defined 3D spheroids with no proper spheroidal boundary (Figure 2G). We performed a dot blot analysis on lysates derived from CAL27 spheroids to assess BRCA1 and RAD17 protein levels upon miR-205-5p depletion (Figure 2B).

We observed a significant increase in the expression of BRCA1 and RAD17 protein levels paired with a decrease in the expression of the proliferation marker Ki67 when compared to the cells transduced with LNA control oligos (Figure 2H).

Taken together, these findings show that aberrant expression of miR-205-5p contributes to the tumoral phenotype of HNSCC cells.

### 2.3. MiR-205-5p Expression Impairs DNA Repair Activity

BRCA1 and RAD17 proteins are deeply involved in double-strand break (DSB) repair and checkpoint activation [1]. One important pathway in cancer cells responsible for the repair of DSBs is non-homologous end-joining (NHEJ), an intrinsically error-prone mechanism as it does not require the presence of an intact DNA template [1]. Since these two main players of DNA repair were shown to be targets of miR-205-5p above, we aimed to test whether the expression of miR-205-5p had an impact on DNA repair activity of A253 head and neck cells exhibiting low miR-205-5p levels (Figure 3A). Comet assays revealed that cells overexpressing miR-205-5p had a greater amount of DNA damage, expressed as the percentage of DNA in the tail of the comets compared to control cells (Figure 3A). We also observed a significant increase in phosphorylated H2AX^Ser139^ in cells overexpressing miR-205-5p (Figure 3B). Therefore, the comet assay performed on CAL27 cells depleted in miR-205-5p showed a reduction in DNA damage (Figure 3C) pairing with a decrease in the phosphorylation state of H2AX^Ser139^ (Figure 3D).

These results suggested that miR-205-5p expression in cancer cells could significantly increase the number of chromosome breaks downregulating DNA repair activity. 

To assess whether endogenous miR-205-5p expression could hinder the DNA repair process, we studied the role of miR-205-5p in DNA double-strand break (DSB) repair. The pUC19 vector was cut with *Hind*III to create 5’-cohesive ends (Figure 3E, lane 1), and used as a substrate. Incubation of the 5’-cohesive-ended linear DNA (2.6 kb) with the cellular extracts derived from CAL27 cells depleted of miR-205 expression resulted in higher DNA repair activity than the incubation with control extracts (LNA control, Figure 3E, lane 2). This was proven by the increased formation of di-oligomers (D) and tri-oligomers (T) and the decrease in linearized forms (L) (Figure 3E, lane 3).

We employed an additional quantitative evaluation of luciferase activity following ligation of a *Firefly luciferase*-carrying vector [20] which was used to study the impact of miR-205-5p expression on the inhibition of DNA repair. In this DNA repair assay, *pSI-CHECK2* plasmid, linearized by *ApaI* cutting its *Firefly luciferase* gene (Figure 3F), was co-transfected with LNA miR-205-5p or LNA control oligos in CAL27 cells (Figure 3G). Endogenous end-joining activity, resulting in the reconstitution of the luciferase gene, was detected as luciferase activity (Figure 3G). We observed that miR-205-5p depletion significantly rescued the luciferase activity (LNA miR-205-5p vs. LNA control; Figure 3G). Accordingly, BRCA1 and RAD17 protein expression evaluated in the same protein extracts was upregulated (Figure 3H). Of note, we observed that the rescue due to miR-205-5p was diminished by the concomitant interference of either RAD17 (siRAD17) or BRCA1 (siBRCA1) with the concomitant upregulation of two markers of DNA damage, 53BP1 and phosphorylated H2AX^Ser139^ (Figure 3I). The repair activity was still inhibited in control (siGFP/LNA-contr), siBRCA1, and siRAD17 CAL27 cells (Figure 3I).

Altogether, these findings strongly suggest that miR-205-5p significantly impacts the DNA repair ability of head and neck cancer cells through the repression of *BRCA1* and *RAD17* gene expression.

### 2.4. MiR-205-5p Involvement in Genomic Instability

Phosphorylation of H2AX is a well-known marker of double-strand breaks (DSBs) and stalled replication forks. In cancer cells subjected to replicative and oxidative stress, p-H2AX signals are evident in the nucleus in a diffuse and uniform manner [25]. To corroborate the evidence previously showing where miR-205-5p expression caused the increase in phosphorylated H2AX^Ser139^ pairing with a highly damaged DNA (Figure 3B,D), we performed immunofluorescence (IF) staining with anti-P-H2AX^Ser139^ antibody in H1299 cells, a p53-null cell line expressing almost undetectable levels of miR-205-5p, transfected with mimic miR-205-5p or mimic control (Figure 4A). The ectopic expression of miR-205-5p showed a significant increase in P-H2AX signal in the nuclei of cells (Figure 4A).

Micronuclei (MN) spontaneously occur in mammalian cells and are formed in anaphase when chromosome fragments or whole chromosomes fail to segregate into daughter cells [25]. To measure the genomic instability and genotoxic stress, the MN count is widely used. The increase in MN is commonly observed in cells bearing intrinsic genomic instability as in cancer cells and in cells exposed to genotoxic events [25]. According to the hypothesis that miR-205-5p plays a role in tumoral genomic instability, we observed that the ectopic expression of miR-205-5p caused a dramatic increase in the frequency of MN as highlighted by yellow arrows in the Hoechst DNA staining (Figure 4B).

Furthermore, we adopted the IF analysis to study the MN expressing phosphorylated H2AX^Ser139^. We designated those MN coated with P-H2AX signals as MN-P-H2AX (+) (Figure 4C; yellow arrow) and others as MN-P-H2AX (−) (Figure 4C; red arrow). Interestingly, we found that about 60% of MN induced by miR-205-5p expression were MN-P-H2AX (+) compared to the MN present in control cells (Figure 4D). We speculated that the formation of MN-P-H2AX (+) could be associated with DSBs arising from DNA replication caused by the aberrant expression of miR-205-5p.

Previously, it was found that 53BP1-positive punctuated staining usually co-localizes with other DNA damage repair factors, and it is present in preneoplastic and tumoral tissues in vivo [26]. Such 53BP1 staining represents endogenous DNA damage resulting from replicative and oxidative stress, dysfunctional telomeres, and genomic instability of malignant cells, correlating with the sites of incomplete DNA DSB repair [26,27]. Here, 53BP1 foci were present at the basal level in control cancer cells but in significant larger quantities in cells transfected with mimic miR-205 (Appendix A), corroborating the reported prevalence of P-H2AX foci in mimic-miR-205 cells (Figure 4A).

It was shown that mutant p53 proteins promoted aneuploidy, aberrant accumulation of DSBs during cell divisions, centrosome abnormalities, and an increased frequency of aberrant mitosis [20,28,29]. Previously, we assessed that the mutp53-dependent high expression of miR-205-5p played a role in the S-G2 phase that paired with the higher proliferation rate compared to cells depleted of miR-205-5p (Figure 2A,B). Here, we observed that the ectopic expression of miR-205 determined the significant appearance of cells showing severe defects in the cellular division as asymmetric anaphase, chromosome breaks, nuclear and chromosomal bridges, and blebs (Figure 4E,F). These findings agree with the high growth rate (Figure 2A) and the reduced cell-cycle DSB checkpoint activity in the S/G2 phase documented above (Figure 2B).

All these data supported the notion that miR-205-5p counteracted repair processes in front of an induction of cell proliferation rate generating de novo chromosome breaks and/or enhanced DNA damage.

### 2.5. Intratumoral Injections of Mir-205-5p Inhibitor Downsize Established HNSCC Xenografts

To investigate whether the role of miR-205-5p is relevant in inhibiting in vivo cell proliferation and in repressing DNA repair gene expression, we injected HNSCC CAL27 cells to establish subcutaneous tumors in Balb/C nude mice. When tumors with an average volume of 50 mm^3^ were formed, mice were allocated into two groups and either treated with highly stable LNA™-enhanced microRNA inhibitors specific for in vivo miR-205-5p inhibition or a negative LNA control miR (Figure 5A). The inoculations were performed by intra-tumorally administering LNA miRNAs every three days. As shown in Figure 5B, local delivery of LNA miR-205 significantly reduced tumor growth compared to that of control mice (*p* < 0.05 for time points with asterisks), although the tumors progressively grew in all groups during the course of the experiment (Figure 5B).

Immunohistochemistry analysis (IHC) of tumoral tissues of both mice groups revealed lower expression of Ki67 in mice treated with LNA miR-205 injections when compared to the LNA control (Figure 5C). This result paired with the previous findings that showed miR-205 depletion reduced proliferation of CAL27 cells (Figure 2A,B). The distribution of tumor weight and volume, at the time of the last injection among the two mice groups, is shown in Figure 5D,E. Furthermore, miR-205-5p expression was analyzed in both miR-205-treated and control tumors after the sacrifice (Figure 5F).

To determine the correlation between the expression of DNA repair genes and the inhibition of tumor growth by miR-205-5p injection, we stained tumoral tissues of the sacrificed mice for *BRCA1* and *RAD17* expression. We found that *BRCA1* and *RAD17* expression was higher in tumoral tissues of mice treated with LNA miR-205 injections than those with LNA control (Figure 5G and Figure 5H, respectively). These results agreed with the data obtained in CAL27 cells transfected with LNA-miR205-5p (Figure 1I).

### 2.6. Tumoral and Histologically Tumor-Free Peritumoral Tissue Associate with a High Expression of Mir-205-5p in HNSCC Patients

We previously published that, in a cohort of 121 HNSCC patients collected at the *IRCSS* Regina Elena National Cancer Institute (Scientific Institutes of Hospitalization and Care), *TP53* mutations were associated with a shorter recurrence-free survival (RFS) [17,19]. Here, we observed that a high expression of miR-205-5p appeared to be associated with a shorter RFS of HNSCC patients (Figure 6A) [19]. Therefore, to evaluate miR-205-5p expression on matched tumoral (T), peritumoral (PT), and normal (N) tissues of this cohort, we randomly considered 63 HNSCC out of the 121 tissues samples [18,19]. Thirty-two sample patients carried *TP53* missense mutations, and 31 exhibited wild-type *TP53* (tumor and non-tumoral matched tissues for each patient) (Appendix A) [20]. Through the Taq-Man PCR assay, we assessed that miR-205-5p was upregulated in tumor tissues compared with matched non-tumoral tissues (Figure 6B). Notably, miR-205-5p expression was higher in the *TP53*-mutated patients (Appendix A) than in those with intact *TP53* (Appendix A). The analysis of tumoral tissues from HNSCC TCGA confirmed what we found for miR-205-5p expression compared to the *TP53* status (Figure 6C; Appendix A).

We also looked at *BRCA1* and *RAD17* mRNA expression in HNSCC TCGA. We found that their expression was significantly and positively correlated (Appendix A) and that a low expression of the *BRCA1*/*RAD17* signature was associated with short recurrence in HNSCC patients (Figure 6D). Notably, the *BRCA1*/*RAD17* signature was specifically lower in the subgroups of *TP53*-mutated patients than in that with intact *TP53* (Appendix A). We also observed that *BRCA1* and *RAD17* expression significantly anti-correlated with that of miR-205-5p, with significant anti-correlation in *TP53*-mutated patients (Appendix A).

Approximately 60% of HNSCC patients have a locally relapsed tumor, which is the most common cause of death in HNSCC patients [30]. The “field cancerization” hypothesis provides a mechanism for the local recurrence. It was extensively shown that mucosa of HNSCC patients from which the primary tumor arises are apparently healthy when analyzed from a histological point of view but present several genetic modifications when analyzed at a molecular level [31]. This hypothesis is supported by the fact that clonal relationships between the primary tumor and the tumor-adjacent premalignant epithelium supports were highlighted [32,33,34]. We recently reported that peritumoral tissues taken at 1 cm from the tumoral lesions appeared histologically undistinguishable from matched non-tumoral tissues [30]. Notably, we previously reported that the expression of specific miRNAs was already altered in the peritumoral tissues and was able to predict local recurrence [30].

Here, we wondered whether miR-205-5p from HNSCC patients might predict local recurrence. MiRNA expression profiles of 66 matched tumoral, peritumoral, and non-tumoral patient tissues from the HNSCC patients enrolled at our Institute revealed that miR-205-5p expression was higher in the peritumoral tissues than in the non-tumoral counterparts (Figure 6E). Comparable levels of miR-205-5p were evident in the tumoral and peritumoral tissues, thereby indicating furthermore that tumoral and peritumoral tissues were quite similar regarding the miR-205-5p expression compared to the non-tumoral tissues (Figure 6F). Next, we scored HNSCC patients according to the levels of miRNA-205-5p expression in both peritumoral and matched tumoral tissues. We distinguished three patient groups (high in T/high in PT, low in T/low in PT, other combinations) and evaluated local recurrence-free survival (RFS) of these three groups by using the Kaplan–Meier analysis (Figure 6G). Notably, we observed that patient groups (high in T/high in PT) had a shorter local recurrence-free survival compared with the other groups (Figure 6G).

To study the deregulation of miR-205-5p in peritumoral tissue from a functional perspective, we combined the information of miR-205 expression levels with the gene expression profiles obtained by microarrays previously carried out by 22 tumoral, peritumoral, and non-tumoral matched tissues [30]. Firstly, we identified genes whose expression was significantly modified in peritumoral vs. normal matched samples (*p* < 0.05). Then, we analyzed whether the expression of these genes either positively or negatively correlated with the expression of miR-205-5p. We obtained 372 correlated mRNAs, of which 227 positively and 145 negatively correlated according to the Pearson’s correlation coefficient higher than 0.3 in absolute value (Figure 6H; Appendix A).

Gene set enrichment analysis (GSEA) functional classification of positive correlated mRNAs showed enrichment for genes belonging to various cancer-related pathways, such as the axon guidance involved in metastatization processes, metabolism of insulin, PI3K/Akt, and Ras signaling pathways (Appendix A).

These results indicated that the deregulation of miR-205-5p in a subset of tumor-surrounding mucosa tissues might be relevant for the clinical outcome of the disease.

## 3. Discussion

Genomic instability is a main feature of many types of human cancers including HNSCC. In this study, we showed that the aberrant expression of non-coding factors, such as miR-205-5p, caused inefficient DNA repair, strongly contributing to endogenous DNA damage accumulation in HNSCC. Indeed, miR-205-5p aberrantly regulated DNA repair, consequently shutting down the expression of two important DNA repair genes, *BRCA1* and *RAD17*.

MiR-205 was firstly predicted by computational approaches and, subsequently, its expression was validated in zebrafish and humans [35,36].

MiR-205 appears to have controversial roles in malignancy [37,38]. MiR-205 was found to be either up- or downregulated in breast cancer and may function as either a tumor suppressor or an oncogene. Diverse studies showed that miR-205 can promote tumor initiation, progression, and resistance to anti-tumor therapy [39,40,41,42,43]. Overall, it emerged that the role of miR-205-5p in cancer depends strongly on the cellular context and might likely be specific for a tumor subtype, as well as its cellular origin and stage of tumor progression. Our findings shed light on the roles of miR-205-5p in HNSCC that are characterized by a high degree of intratumoral heterogeneity. This strongly hampers the development of robust prognostic tools with broad clinical utility. High miR-205-5p expression levels in HNSCC patients were previously reported [44,45,46]. We previously documented that miR-205-5p belongs to a signature of 12 miRNAs predicting the risk of local recurrence insurgence, as well as being upregulated by mutp53/NF-Y and mutp53/E2F1 transcriptional competent complexes in HNSCC [18]. Here, we report that high miR-205-5p expression is associated with low recurrence-free survival in HNSCC patients and that its upregulation is significantly related to the subgroup of tumors with *TP53* mutated (Figure 6). 

To date, several target mRNAs ascribed to miR-205 including *PTEN* [39] and *SHIP2* [41], such as tumor suppressors, *HER3* [43], *E2F1*, *E2F5*, and *PKCε* [43], and oncogenes were identified. Here, we unveil two novel miR-205-5p target genes, *RAD17* and *BRCA1*. Both gene products are pivotal players involved in cell-cycle arrest, DNA damage response, and in maintenance of genomic stability. Clinically, the low expression of the *BRCA1*/*RAD17* signature in HNSCC tumor samples was found to be significantly anti-correlated with miR-205-5p expression (Appendix A) and associated with high recurrence (Figure 6D).

El Bezawy et al. considered pancreatic cancer (PC) as a working model characterized by radioresistance and low miR-205 expression levels [47]. MiR-205 reconstitution by an miRNA mimic in PC cell lines bypassed the DNA repair checkpoint, as a consequence of *PKCε* and *ZEB1* inhibition, and proliferated in a deregulated way, showing a significant increase in the radiation response [47]. In the HNSCC model, miR-205-5p has a high and constitutive expression that inhibits *BRCA1* and *RAD17* targets, thus providing a poor repair ability for replicative DNA damage.

Of note, we also provide evidence that aberrant expression of miR-205-5p occurs in matched peritumoral tissues. According to the field cancerization hypothesis, this epithelium appears to be macroscopically non-tumoral and, thus, rather different from tumoral tissues; however, it is characterized by alterations at both epigenetic and genetic layers [30]. Importantly, these fields are often found in the surgical resection margins and, thus, these tissues might be left in patients, causing severe relapse [30,31,32,33,34].

We documented that miR-205-5p expression is higher in tumoral and peritumoral HNSCC tissues than in non-tumoral tissues in HNSCC patients exhibiting reduced local recurrence-free survival (Figure 6E–G). The identification of predictive miRNA expression in peritumoral tissues might be relevant for early detection of minimal residual disease and of pre-cancer molecular alterations implicated in malignant transformation. Our findings highlight miR-205-5p as a candidate non-coding factor whose evaluation at the resection margins could help identify earlier relapsing HNSCC patients. We previously showed that miR-205-5p resides in the context of the *MIR205HG* locus, whose aberrant activation by gain-of-function mutant p53 led to the production of two non-coding factors lncMIR205HG and miR-205-5p, respectively [20].

At the cellular level, the silencing of miR-205-5p was able to rescue the DNA repair process (Figure 3), usually inhibited in cell lines expressing mutp53 proteins [9]. Moreover, miR-205-5p promoted other pro-tumorigenic activities such as increased cell proliferation rate, the ability to form colonies, and cell migration (Figure 2). These pro-tumorigenic activities were also highlighted in vivo (Figure 5B,C).

Collectively, these results suggest a model in which gain-of-function mutant p53 protein acts aberrantly, at least, at two different regulation levels to maintain genomic instability in cancer cells; it is able to transcriptionally inhibit *BRCA1* and *RAD17* gene expression [9] and to induce miR-205-5p [20] that targets *BRCA1* and *RAD17* (Figure 6I). This increases defects in DDR and DSB repair activities, causing detrimental chromosomal abnormalities (Figure 4). The accumulation of unrepaired DNA and secondary insults might contribute to establishing a cellular environment fostering disease progression and metastatization affecting multiple signaling pathways, leading to genomic instability. We speculate that this mechanism may also be conserved in other cancers expressing high levels of miR-205-5p and mutant p53 protein (e.g., ovarian carcinoma, triple-negative breast cancers), characterized by high genomic instability.

The depicted link between miRNAs and genomic instability, at least in the context of HNSCC tumors, could have important implications in being translated into the clinical practice for predicting and monitoring relapse. Furthermore, the knowledge of the miRNA dysregulation in human cancers might provide new molecular insights either for designing novel therapeutic approaches or repurposing existing drugs.

## 4. Materials and Methods

### 4.1. Cells

Cells were maintained in culture for no more than six passages. All cell lines were tested by PCR/IF for *Mycoplasma* presence. Head and neck CAL27 (mutp53H193L), A253 (p53-null), and lung H1299 (p53-null) cancer cell lines were cultured in RPMI medium (Invitrogen, Carlsbad, CA, USA); all media were supplemented with 10% (*v*/*v*) FBS (Invitrogen, Carlsbad, CA, USA). All cell lines were purchased from ATCC and were authenticated by STR (Short Tandem Repeats) genotyping with the Promega PowerPlex® 1.2 system and the Applied Biosystems Genotyper 2.0 software for analysis of the amplicons.

### 4.2. Comet Assays

In order to quantify the DNA damage, single-cell gel electrophoresis was performed with a Comet Assay kit following the manufacturer’s instructions (Trevigen Gaithersburg, MD, USA). Cells were detached with trypsin and embedded in 1% low-melting agarose and spread onto microscopy slides coated with 1% agarose. Cells were lysed in the alkaline lysis solution that detects DNA single-strand breaks, double-strand breaks, and alkali-labile lesions, and then run in running solution (300 mM NaOH, 1mM EDTA) for 30 min at 1 V/cm and about 300 mA. DNA was dried with 70% ethanol, stained with DAPI, and mounted with Vectashield (Vector Labs, Burlingame, CA, USA). Pictures were taken using an Axiovert 200M microscope and Axiovision acquisition program (Zeiss). At least 200 cells were scored for each slide. A software tool was used to provide an automated analysis of comet assay images (OpenCOMET; www.opencomet.org).

### 4.3. 3D Spheroid Formation

In order to determine the best cell concentration for long-term in vitro culture, CAL27 cells with an initial concentration of 1, 2, 4, and 10  ×  10^6^ cells/mL were mixed into Alginate Hydrogel solution and plated in 96-well plates; then, a matrix was crosslinked with Crosslinking Solution at room temperature following the manufacturer’s instructions (3D Cell Culture Matrix Alginate Hydrogel Kit n° K517-100, BioVision Milpitas, CA). Then, the spheroids were cultured at 37 °C with 5% CO_2_ for 10–15 days. The culture medium was replaced every three days. We analyzed the spheroid size using a confocal inverted microscope (Olympus Life Science). To analyze RNA and protein expression, a Matrix Dissociation Buffer (n° M1090 BioVision Milpitas, CA) was used to recover the CAL27 spheroids from the matrix.

### 4.4. NHEJ DNA Repair In Vitro Assay

The DNA repair in vitro assay was performed as previously described [20]. Linearized pUC19 plasmid DNA was digested with *Hin*dIII restriction enzyme to produce complementary ends. DNA was purified with a DNA extraction kit (Qiagen) from the agarose gel. Cell extracts were prepared using hypotonic buffer (10mM HEPES–KOH pH 7.9; 1.5 mM MgCl_2_; 10 mM KCl; 0.5 mM dithiothreitol; 0.2 mM phenylmethylsulfonyl fluoride) and three cycles of freeze–thawing, followed by centrifugation at 13,000 rpm. The repair assay was performed in 50 μL of reaction buffer (50 mM Tris pH 8.0, 5 mM MgCl_2_, 1 mM ATP, 1 mM DTT, 5% polyethyleneglycol 8000, protease inhibitor cocktail) with 200 ng of substrate DNA and 20 ug of protein, incubated at 17 °C for 16 h. The repair reaction was determined by adding 0.4% SDS and incubating at 65 °C for 15 min. DNA was recovered by extraction with phenol–chloroform (1:1 *v*/*v*) and ethanol precipitation, and repair products were identified by 1% agarose electrophoresis. 

### 4.5. DNA Repair Assay with Luciferase Reporter Gene

Cells (2 × 10^5^) were seeded into six- or 12-well culture plates and transiently transfected with 100 ng of reporter constructs as described in the Appendix A. Then, psi-CHEK2^TM^ was cut with an *Apa*I restriction enzyme in the firefly luciferase domain. The linearized vector was transfected as previously reported. Additionally, 1/10 of CMV-*Renilla* plasmid as an internal control was co-transfected for transfection efficiency. Firefly luciferase activity was measured with a Dual Luciferase Assay Kit (Promega) 48 h after transfection and normalized with a *Renilla* luciferase reference plasmid. Reporter assays were carried out in quadruplicate, and the mean ± SD was reported. Statistical significance was analyzed by the unpaired Student *t*-test.

### 4.6. HNSCC Xenografts

For miR-205-5p injection experiment, 10^6^ CAL27 cells in 30% matrigel were subcutaneously injected into immunodeficient Balb/C mice. When tumors reached an average volume of 50 mm^3^, 50 μL of synthetic miRNA (miR-205-5p inhibitor n° YI04101508; LNA negative control A n° YI00199006 were purchased by Exiqon-Qiagen) complexed with Invivofectamine® 3.0 transfection reagent (ThermoFisher, Life tech.) was delivered intratumorally in four-day intervals. For each injection, 7 μg miRNA was complexed with Invivofectamine® 3.0 transfection reagent in 50 mL of PBS. At the end of the treatments, animals were sacrificed in accordance with standard protocols, while tumors were collected and prepared for histology and RNA isolation. All the procedures involving animals and their care were performed by the company Biogem s.c.ar.l of Ariano Irpino (Italy) in line with the Italian legislation and the guidelines of ISO 9001:2008 second edition (10 June 2009).

### 4.7. Head and Neck Tumor Tissue Samples

The head and neck squamous cell carcinoma patients with the description of *TP53* status and methods of RNA extraction for each patient were previously described [17,19] (Appendix A). The Scientific Ethics Committee of the Regina Elena National Cancer Institute approved the study protocol (CE/379/08). The direct sequencing of exons 2–11 of the *TP53* gene was employed to detect the mutations. MiR-205-5p expression was analyzed by RT-qPCR in a group of 63 samples and their non-tumoral counterparts. Non-tumoral samples were collected from surgery resection margins from each patient and were all histologically checked for the absence of tumor cells [17]. Non-tumoral samples were also subjected to *TP53* sequencing and were negative for mutations.

### 4.8. Bioinformatic Analysis

The mature miR-205-5p sequence was taken from the miRBase database [21]. We searched for miR-205-5p “seed sequence” on *BRCA1* and *RAD17* mRNA using MiRWALK 3.0 [22,48].

### 4.9. Statistical Analysis

Data were presented as means ± SD or standard error of the mean (SEM), derived from at least three independent experiments. ANOVA analysis was performed by the R project tool. Student’s *t*-test (two-tailed) was also conducted. A *p*-value <0.05 was indicated as statistically significant.

### 4.10. Ethical Approval and Consent to Participate

All procedures involving animals and their care were performed by the Biogem s.c.ar.l company of Ariano Irpino (Italy) in accordance with the Italian legislation and guidelines of ISO 9001:2008 second edition (10 June 2009).

## 5. Conclusions

In this study, we identified a novel molecular crosstalk between mutant p53 protein and miR-205-5p in HNSCC. We found that BRCA1 and RAD17, two proteins involved in the DNA repair process, are targets of miR-205-5p in HNSCC cellular models and in patients, determining the progressive accumulation of unrepaired DNA. Higher expression levels of miR-205-5p were detected in tumoral and peritumoral HNSCC tissues than in non-tumoral tissues in HNSCC. The identification of miR-205-5p in peritumoral tissues could represent an important step for early detection of minimal residual disease implicated in tumoral transformation.

## Figures and Tables

**Figure 1 cancers-11-01347-f001:**
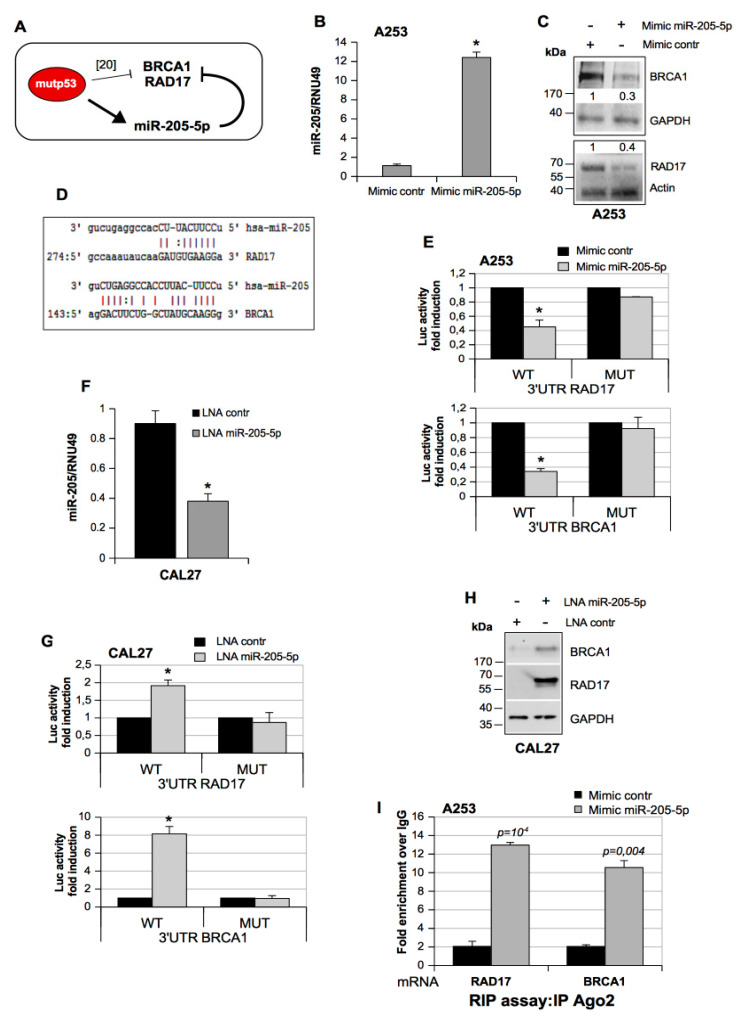
MicroRNA (MiR)-205-5p targets *BRCA1* and *RAD17* DNA repair genes. (**A**) Schematic model representing the different levels of *BRCA1* and *RAD17* gene expression inhibition orchestrated by the mutant p53 protein (mutp53)/miR-205-5p axis. (**B**) A253 cells transfected with mimic control and mimic miR-205-5p oligos are analyzed for miR-205-5p expression using a Taq-Man assay. The RNA levels were normalized versus *RNU49*. Bars represent the mean ± SD from three biological replicates. The *p*-values were calculated using a two-tailed *t*-test. * *p* < 0.05. (**C**) Representative Western blotting of BRCA1, RAD17, and GAPDH (loading control) protein expression in A253 cells treated as described in (B). (**D**) Representation of the pairing between miR-205-5p and the 3’ UTR (untraslated region) of its putative messenger RNA (mRNA) targets, *BRCA1* and *RAD17* mRNAs. (**E**) Activity for *RAD17* 3’ UTR and *BRCA1* 3’ UTR luciferase reporter vectors and mutated *RAD17* 3’ UTR and *BRCA1* 3’ UTR were determined after 48 h of transfection with mimic control and mimic miR-205-5p oligos in A253 cells. The results derived from three independent experiments. Bars represent the mean ± SD from three biological replicates. The *p*-values were calculated with a two-tailed *t*-test. Significant *p*-values are indicated as * *p* < 0.05. (**F**) MiR-205-5p expression was analyzed by a Taq-Man assay from CAL27 cells transfected with LNA (locked nucleic acid) control and LNA miR-205-5p oligos. The expression was normalized versus RNU49. Bars represent the mean ± SD from three biological replicates. The *p*-values were calculated with a two-tailed *t*-test. * *p* < 0.05. (**G**) Luciferase activity for *RAD17*-3’ UTR-pGL3 and *BRCA1*-3’ UTR-pGL3 plasmids and mutated *RAD17* 3’ UTR and *BRCA1* was determined after 48 h of transfection with LNA control and LNA miR-205-5p oligos in CAL27 cells. The results derived from three independent experiments. Bars represent the mean ± SD from three biological replicates. The *p*-values were calculated with a two-tailed *t*-test. Significant *p*-values are indicated. * *p* < 0.05. (**H**) Western blotting of BRCA1, RAD17, and GAPDH (loading control) protein expression in CAL27 cells transfected with either LNA control or LNA miR-205-5p. (**I**). RIP (RNA Immuno Precipitation) assays performed in control and miR-205-5p overexpressing A253 cells using antibodies directed to Ago2 protein. RNA abundance of Ago2 was evaluated by RT-qPCR and normalized over GAPDH mRNA and over IgG as negative control; a representative experiment of a biological duplicate is shown.

**Figure 2 cancers-11-01347-f002:**
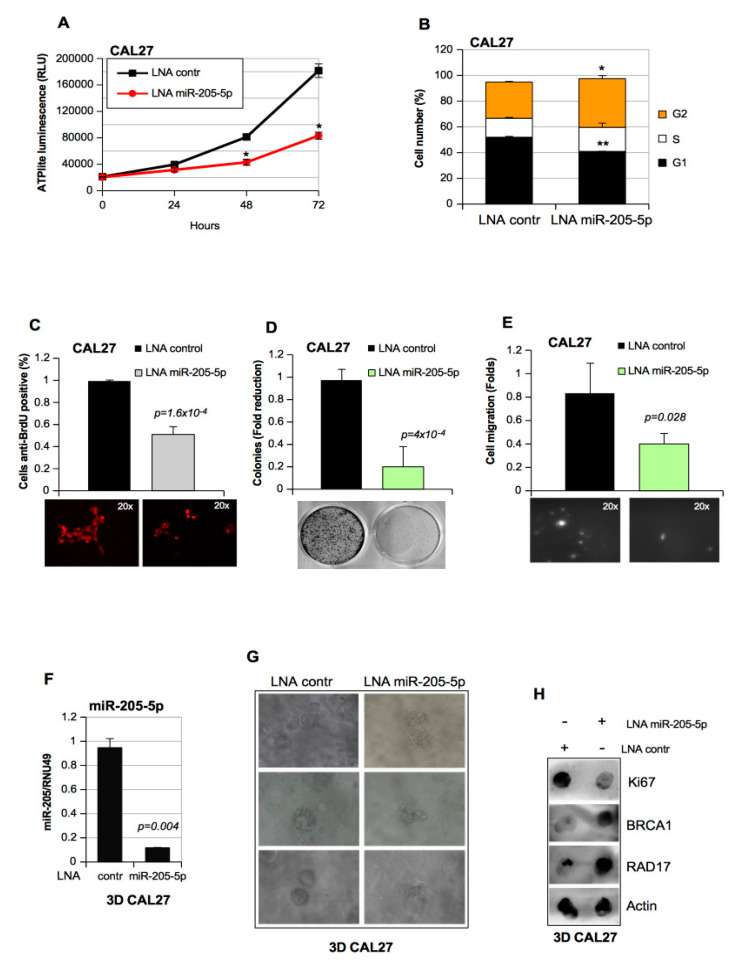
Inhibition of miR-205-5p impairs the tumoral features of head and neck squamous cell carcinoma (HNSCC) cell lines. (**A**) CAL27 cells transiently transfected with LNA miR-205-5p or LNA control at the indicate time points were subjected to label-free assays, and the ATP levels are expressed as relative light units (RLUs). Values are the mean ± SD obtained from two separate experiments, each plated in quintuplicate. A two-tailed *t*-test was used to calculate the *p*-values: * *p* < 0.05 versus the control sample. (**B**) CAL27 cells transfected for 48 h with LNA control and LNA miR-205-5p oligos were incubated with PI staining solution and analyzed by flow cytometry. The percentage of cells (%) in different phases are shown. Data are presented as the mean ± SD of three independent experiments; * *p* < 10^−5^, ** *p* < 10^−8^. (**C**) Cells treated the same as in (B) were incubated for 20 min with BrdU. The BrdU-positive cells were scored by immunofluorescence staining, and total cells were counted under transmission light. Data are expressed as the mean percentage of BrdU-positive cells ± SD. (**D**) CAL27 cells transfected with LNA control and LNA miR-205-5p were subjected to the colony formation assay. All the values are the mean ± SD of six replicates from three independent experiments. Analysis of the colonies was performed using the ImageJ software. (**E**) Representative images of CAL27 cells from the transwell migration assay by the Boyden chamber. Data represent the mean ± SD from three biological replicates; each point was repeated in technical quadruplicates. The *p*-values were calculated with a two-tailed *t*-test. All *p*-values were calculated with a two-tailed *t*-test and were higher than the significance level of 0.05. (**F**) RNA from CAL27 3D cells transfected with LNA control and LNA miR-205-5p oligos was evaluated for miR-205-5p expression by a Taq-Man assay. The expression was normalized using RNU49. Bars represent the mean ± SD from three biological replicates. The *p*-values were calculated with a two-tailed *t*-test. (**G**) Bright-field representative images of CAL27 transfected for 48 h with the indicated LNA oligos and grown for 10 days in a three-dimensional (3D) cell culture using a matrix alginate hydrogel (32× objective). (**H**) Representative image of dot blot assay: 2 ug of whole-protein lysates extracted from CAL27 3D culture described in (F) and (G) were dotted onto a nitrocellulose membrane and immunostained for BRCA1, RAD17, Ki67, and Actin (loading control) protein expression.

**Figure 3 cancers-11-01347-f003:**
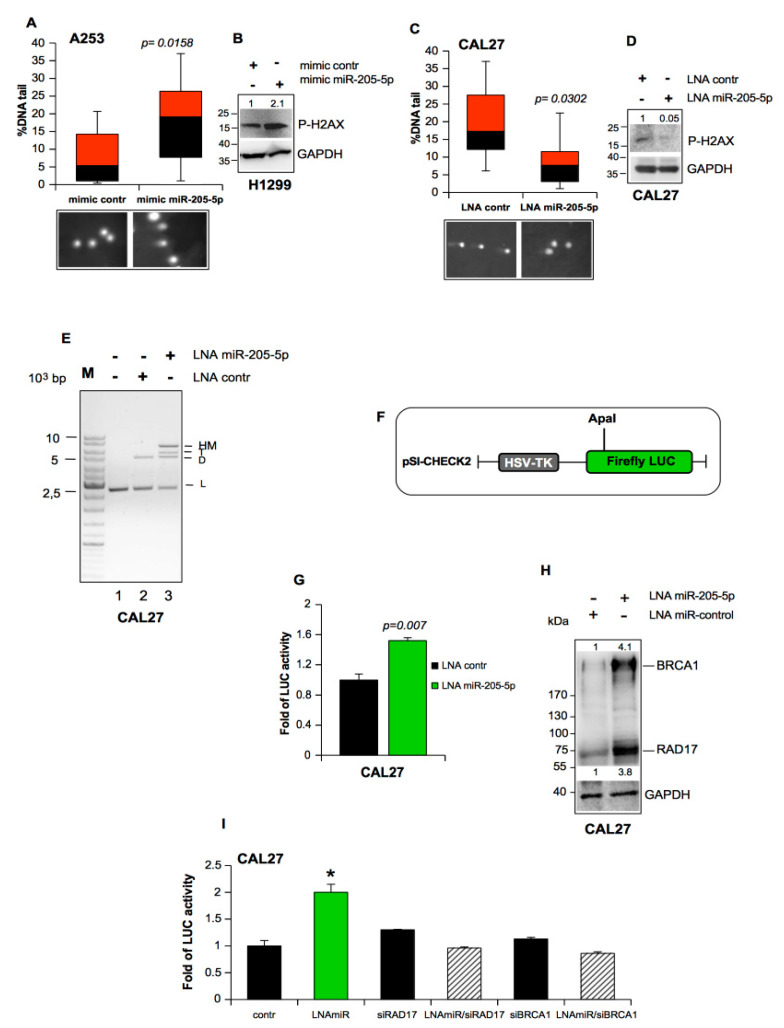
MiR-205-5p expression impairs DNA repair activity. (**A**) H1299 cell lines transfected with mimic control and mimic miR-205-5p were analyzed for the amount of damaged DNA by comet assay using the alkali method. The extent of DNA damage is related to the amount of DNA in the tail. The percentage of DNA in the tail is reported as a box-plot in the higher panels of each figure. About 100 cells were counted for each sample. In all box-plots, the black box is the 25th percentile (first quartile) and the gray box is the 75th percentile (third quartile), while the black band is the median (second quartile). The *p*-values were calculated with a two-tailed *t*-test. Statistically significant results had a *p*-value <0.05. (**B**) Representative Western blotting of P-H2AX and GAPDH (loading control) protein expression in H1299 cells transfected as described in (A). (**C**) CAL27 cells transfected with LNA control and LNA miR-205-5p were analyzed for the amount of DNA damaged by comet assay using the alkali method. Analysis was performed as described in (A). (**D**) Representative immunoblotting analysis of P-H2AX and GAPDH (loading control) protein expression in CAL27 cells transfected as described in (C). (**E**) DNA repair products after 16 h of incubation at 17 °C with cellular extracts of CAL27 cells transiently transfected with LNA control (lane 2) and LNA miR-205-5p (lane 3) oligos. Lane 1: reaction buffer mixture with *Hind*III-cut plasmid. DNA bands include the following: L, linear DNA; D, dimer; T, trimer; HM, high multimer. Numbers indicate the lanes. M is the DNA ladder. (**F** and **G**) CAL27 cells were transiently transfected with *pSI-CHECK2* plasmid cut using *ApaI* (F) and with either LNA control or LNA miR-205-5p oligos as indicated in the figures (G). Then, 48 h after transfection, the cells were harvested, and the DNA repair was assessed measured by firefly luciferase activity. *Columns*, means from two independent assays each done in triplicate; *bars*, SD. The *p*-values were calculated with a two-tailed *t*-test. Statistically significant results had a *p*-value <0.05. (**H**) Representative immunoblotting analysis of BRCA1, RAD17, and GAPDH (loading control) protein expression in CAL27 protein cell extracts used in (G). Anti-BRCA1 and anti-RAD17 antibodies were used in the same hybridization solution. (**I**) CAL27 cells were transiently transfected with ApaI-linearized pSI-CHECK2 vector and LNA miR-205-5p (LNAmiR), siRAD17, and siBRCA1 oligos in different combinations as indicated in the figures (I). The control sample (contr) involved CAL27 cells transfected with LNA control and siGFP oligos. Then, 48 h after transfection, the cells were harvested, and the functional changes in DNA repair were assessed measuring the firefly luciferase activity. The *p*-values were calculated with a two-tailed *t*-test. Statistically significant results had a * *p* < 0.05.

**Figure 4 cancers-11-01347-f004:**
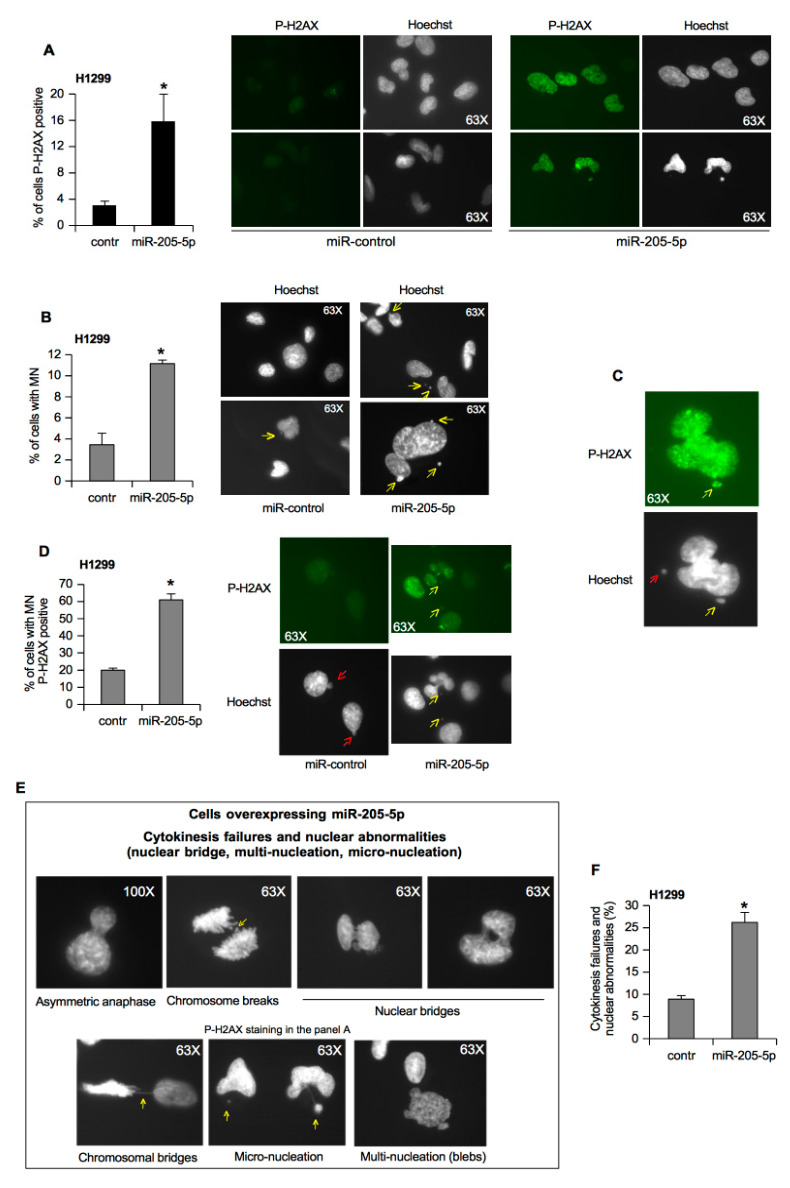
MiR-205-5p involvement in genomic instability. (**A**) H1299 cells were transfected with mimic miR-205-5p and mimic control oligos. Subsequently, cells were stained for P-H2AX 48 h after miR-205-5p and mimic control oligo overexpression. Representative images and quantification of unrepaired double-strand breaks (DSBs) are shown. Hoechst was used for nuclear staining. The *p*-values were calculated with a two-tailed *t*-test. Statistically significant results had a *p*-value <0.02 (*). Data represent the mean ± SD from three biological replicates. (**B**) H1299 cells overexpressing miR-205-5p and a mimic control were observed on the basis of micronuclei (yellow arrows). The percentage of 500 cells scored by each point is reported. The *p*-values were calculated with a two-tailed *t*-test. Statistically significant results had a *p*-value <0.02 (*). Data represent the mean ± SD from three biological replicates. (**C**) Examples of micronuclei (MN)-P-H2AX (+) (yellow arrow) and MN-P-H2AX (−) (red arrow) in H1299 cells. Cells were grown on coverslips in six-well plates 48 h before they were processed for immunofluorescence staining with anti-P-H2AX antibody. (**D**) H1299 cells overexpressing miR-205-5p and a mimic control are represented on the basis of MN-P-H2AX (+) (yellow arrow) and MN-P-H2AX (−) (red arrow). The percentage of 500 cells scored by each point is reported. The *p*-values were calculated with a two-tailed *t*-test. Statistically significant results had a *p*-value <0.05 (*). Data represent the mean ± SD from three biological replicates. (**E**) Chromosomal abnormalities in H1299 cells overexpressing miR-205-5p. The arrows in different representative images indicate conspicuous chromatin bridges between cells in anaphase or telophase, loss of chromosomal material, nuclear bridges, and blebs. (**F**) The percentage of 500 cells scored by three biological replicates of H1299 cells overexpressing miR-205-5p is reported. The *p*-values were calculated with a two-tailed *t*-test. Statistically significant results had a *p*-value <0.02 (*). Data represent the mean ± SD.

**Figure 5 cancers-11-01347-f005:**
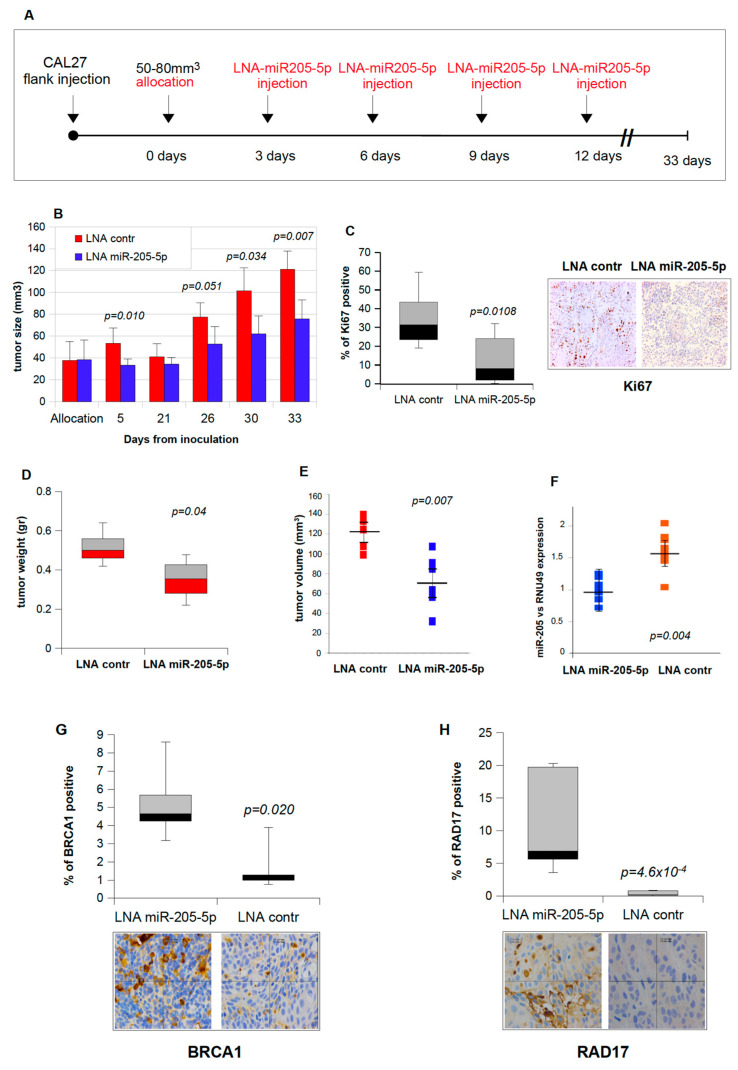
Intratumoral injections of miR-205-5p inhibitor downsize established HNSCC xenografts. (**A**) Schematic overview of in vivo experiments performed on orthotopic HNSCC. The tumor growth was monitored with the aid of a caliper. Mice assigned to the treatment and control groups were treated three times with miR-205-5p inhibitors or corresponding control oligonucleotides every three days; *N* = 6 mice in each group. In each panel, *p*-values were calculated by a two-sample *t*-test. (**B**) A total of 10^6^ CAL27 cells in 30% matrigel were subcutaneously injected into immunodeficient Balb/C mice. On days 3, 6, 9, and 12 following mice allocation, synthetic LNA miR-205-5p inhibitor or control miRNAs conjugated with the Invivofectamine® 3.0 transfection reagent were intratumorally delivered into groups of six animals. Caliper measurements were taken to determine the length and width of each tumor and to calculate total tumor volumes. (**C**) Immunohistochemical analysis of Ki67 protein expression was analyzed in six LNA control- and LNA miR-205-5p-treated mice. Representative images and the relative quantification of ki67 positive cells are shown. (**D**) Tumor weight of the excised tumors measured at the end of the experiment. (**E**) Tumor volumes measured at the end of the experiment. (**F**) Relative miR-205-5p expression in CAL27 tumors. Total RNA was extracted from tumors harvested at the end of the experiment, and RT-qPCR was performed using a specific probe for miR-205-5p (Taq-Man assay). The normalization was carried out using RNU49 throughout the standard curve method. The *p*-values were calculated by a two-sample *t*-test; significant results are marked by a *p*-value <0.05. (**G**,**H**) Immunohistochemistry on tumors treated with LNA control and LNA miR-205-5p. Sections from each mouse were incubated with an anti-BRCA1 antibody (**G**) and anti-RAD17 antibody (**H**). Representative fields are shown.

**Figure 6 cancers-11-01347-f006:**
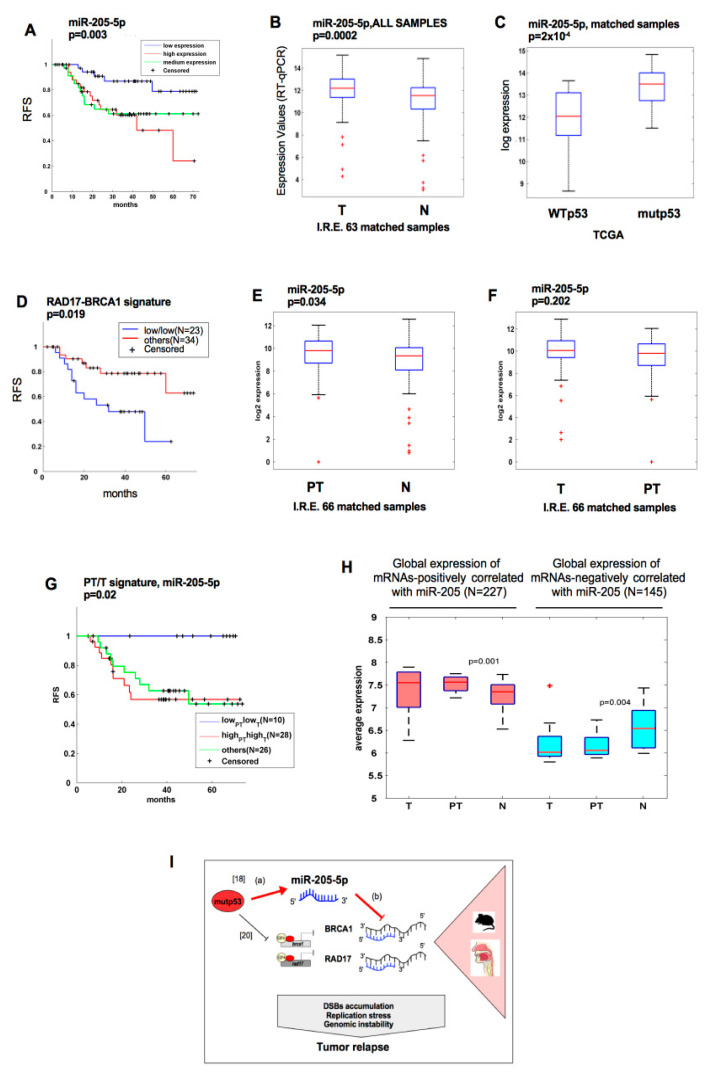
Tumoral and histologically tumor-free peritumoral tissue is associated with a high expression of miR-205-5p in HNSCC patients. (**A**) Kaplan–Meier survival curve and log-rank test for 121 HNSCC patients showing recurrence-free survival (RFS). High and medium expression of miR-205-5p was associated with significantly decreased RFS (*p* = 0.003) in HNSCC. (**B**) Box-plot analysis representing the association of miR-205-5p expression levels with the tumoral (T) and non-tumoral (N) tissues from the HNSCC cohort of *IRCSS* Regina Elena National Cancer Institute of Rome. The expression of miR-205-5p was measured by RT-qPCR in complementary DNA (cDNA) obtained from each non-tumoral and matched tumoral tissues. The 63 matched samples include tumors carrying missense mutations in *TP53* (mutp53, *n* = 32) and tumors with wild-type *TP53* (WTp53, *n* = 31). (**C**) Box-plot analysis representing the association of miR-205-5p expression levels within the following sub-groups of 42 matched samples from TCGA: mutant p53 (tumors carrying mutations in *TP53* (*n* = 31)); wt p53 (tumors with wild-type *TP53* (*n* = 11)). (**D**) Kaplan–Meier survival curve and log-rank test for 57 TCGA HNSCC patients showing recurrence-free survival (RFS). Low expression of the *RAD17*/*BRCA1* signature was associated with significantly decreased RFS (*p* = 0.019) in HNSCC. Box plots showing the miR-205-5p expression differentially expressed between 66 peritumoral (PT) tissues and their normal counterparts (*p* < 0.05) (**E**) and between 66 tumoral tissues and their peritumoral matched tissues (**F**). (**G**) Kaplan–Meier (KM) analysis showed the correlation between miR-205-5p expression and RFS in tumor and peritumoral tissues. KM analyses were performed through creating four groups of patients: patients with high expression of miRNA for both tissues (tumor and peritumor), patients with high expression in tumor and low expression in peritumor and the contrary, and patients with low expression in both tissues. The *p*-value was calculated for the two subgroups expressing high and low levels of miR-205-5p in both tissues (tumor and peritumor). (**H**) Box-plot representing the expression of 227 genes positively (mRNAs-PCm) and 145 genes negatively (mRNAs-NCm) correlated with miR-205-5p expression and differentially expressed between tumoral and peritumoral tissues and their normal counterparts (*p* < 0.05) in 22 tumoral, peritumoral, and non-tumoral matched tissues of the IRE cohort [30]. (**I**) Schematic representation showing two parallel mechanisms through which mutant p53 is able to counteract *BRCA1* and *RAD17* gene expression. (a) MiR-205-5p is induced by mutp53 gain-of-function (GOF) activity in HNSCC cancer cells [18]. (b) High levels of miR-205-5p target *BRCA1* and *RAD17* mRNAs downregulating their protein expression. (c) In parallel, mutp53 forms a complex with the E2F4 repressor factor and, via binding *BRCA1* and *RAD17* promoters onto the E2F4 repressive consensus sequences, is able to inhibit their transcription [20]. All these activities contribute to reinforcing aberrant mechanisms, resulting in an increase in genomic instability of HNSCC tumors.

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
