# Peer review of "The miR-205-5p/BRCA1/RAD17 Axis Promotes Genomic Instability in Head and Neck Squamous Cell Carcinomas"

_cancers, 2019, doi:10.3390/cancers11091347_

Round 1

Reviewer 1 Report

This manuscript is an follow-up of the authors published work on mechanisms of mutp53 mediated cancer progression. Overall appropriate experimental methods were executed with results are conclusive. There is one suggested experiment in the comments below.However, the authors need to significantly improve their english writing and the figure qualities.

Specific comments are given below:

Abstract as well as the entire manuscript needs to be thoroughly revised for typos, grammatical errors and incorrect sentence constructions. Line 25: Define miR Line 44: Too many ‘in’. Please consult an English writing expert before submitting manuscript. Line 29: “miR-205-5p downregulation reduced in vivo tumor growth leading to increased BRCA1 and RAD17 mRNA levels.” This is confusing. Rather write “miR-205-5p downregulation increased BRCA1 and RAD17 mRNA levels leading to reduced in vivo tumor growth”. Line 52-53: “It requires numerous factors including the RAD17, RAD51 recombinases and the breast / ovarian cancer susceptibility gene products, BRCA1 and BRCA2” – this sentence following the prior one suggest that the mentioned factors are involved in DSB repair in an infidel DSB repair pathway, which is misleading. RAD51, BRCA1, and BRCA2 are involved in HR. Line 60: define mutp53 somewhere. Line 66: correct this sentence “In few cases, defects in the Fanconi Anemia pathway are the responsible or mutations in genes related to chromatid cohesion genes.” Line 105-106: correct the sentenc Line 184: This is not a precise statement "The three-dimensional (3D) spheroids mimic the tumor environment in a highly faithful manner" and should be avoided. Could not find BRCA1 and Rad17 in this list presented in Table S1. Present clearer microscopy images in Figure 2C and 2E are not clear. Provide western blot instead of dot blots in 2H. Line 279-280: correct sentence. Figure 4 is very busy. Remove lower magnification images from 4A. The gammaH2AX staining looks mostly pan-nuclear, without distinct foci. Show if miR-205 treated cells are more sensitive to DNA damaging agent or replicative stress induced by hydroxyurea. Improve image quality of Figure 6. Define T, N, PT, etc in legend.  

Author Response

Dear Editor,

we wish to thank you for providing us with the opportunity to revise our manuscript No. CANCERS_574526 accordingly to the reviewers comments. The comments were very helpful for strongly improving our manuscript.

The main corrections are highlighted in the manuscript. Below please find the detailed list of our responses to the reviewers' suggestions/comments.

Reviewer 1

This manuscript is an follow-up of the authors published work on mechanisms of mutp53 mediated cancer progression. Overall appropriate experimental methods were executed with results are conclusive. There is one suggested experiment in the comments below.However, the authors need to significantly improve their english writing and the figure qualities.

Specific comments are given below:

Abstract as well as the entire manuscript needs to be thoroughly revised for typos, grammatical errors and incorrect sentence constructions.

-Line 25: Define miR Line 44: Too many ‘in’. Please consult an English writing expert before submitting manuscript.

-Line 29: “miR-205-5p downregulation reduced in vivo tumor growth leading to increased BRCA1 and RAD17 mRNA levels.” This is confusing. Rather write “miR-205-5p downregulation increased BRCA1 and RAD17 mRNA levels leading to reduced in vivo tumor growth”.

-Line 52-53: “It requires numerous factors including the RAD17, RAD51 recombinases and the breast / ovarian cancer susceptibility gene products, BRCA1 and BRCA2” – this sentence following the prior one suggest that the mentioned factors are involved in DSB repair in an infidel DSB repair pathway, which is misleading. RAD51, BRCA1, and BRCA2 are involved in HR.

-Line 60: define mutp53 somewhere.

-Line 66: correct this sentence “In few cases, defects in the Fanconi Anemia pathway are the responsible or mutations in genes related to chromatid cohesion genes.”

-Line 105-106: correct the sentenc Line 184: This is not a precise statement "The three-dimensional (3D) spheroids mimic the tumor environment in a highly faithful manner" and should be avoided. Could not find BRCA1 and Rad17 in this list presented in Table S1.

- Line 279-280: correct sentence.

Response: We wish to thank the reviewer for pointing out the sentences which needed English style revision. We made an extensive revision of the English language in which the points raised above by the reviewer have been fully addressed .

- Present clearer microscopy images in Figure 2C and 2E are not clear. Provide western blot instead of dot blots in 2H.

Response: We have replaced the images in Figure 2C and E with acquisitions at higher resolution. Regarding Figure 2H, we tried to perform a Western Blotting but unfortunately the total amount of proteins obtained from the 3D cultures was not more than 10ug total by plating 6 wells for each points in the 96-well plates following the BioVision Milpitas kit. We were unable to detect specifically BRCA1 and RAD17 expression which in fact is obtained by loading at least 30-50 ug of total proteins. We therefore opted to perform dot blots from 2 ug of total protein following the methods of these papers: “A 3D human neural cell culture system for modeling Alzheimer’s disease” by Kim YH et al., Nature Protocols (2015) vol. 10, pages 985–1006 ; “p53 Maintains Genomic Stability by Preventing Interference between Transcription and Replication” by Yeo CQX et al., Cell Rep. (2016) Apr 5;15(1):132-146.

Figure 4 is very busy. Remove lower magnification images from 4A.

Response: We have removed the images as indicated by the reviewer.

The gammaH2AX staining looks mostly pan-nuclear, without distinct foci. Show if miR-205 treated cells are more sensitive to DNA damaging agent or replicative stress induced by hydroxyurea.

Response: To address to this comment we added an additional image in the Figure 4A of the P-H2AX staining in which the foci are very evident. It also shows that the cell on the right side exhibits a micronucleus positive for P-H2AX and while that the left side does not. We have consistently found that upon miR-205-5p overexpression, the cells exhibiting the increase of P-H2AX by western blotting have an enormous amount of foci as can be seen from staining in the Figure 4A. Other images with P-H2AX staining are presented in the Figures 4C and D. The image of the Hoechst staining is shown in figure 4E as an example of micro-nucleation.

- Improve image quality of Figure 6. Define T, N, PT, etc in legend.

Response: We have increased the resolution of the figure and we have clarified better the related figure legend.

Reviewer 2 Report

Base on the authors previous findings (Elife. 2015; doi: 10.7554/eLife.05005; Cancer Res. 2007;67:2396-401. 727), this manuscript confirmed and documented that miR-205-5p, a miR highly expressed in Head and Neck Squamous Cell Carcinoma (HNSCC), targeted BRCA1 and RAD17 expression in vitro as well as in vivo. The results are fully support the mechanisms the authors try to pursuit and will be interesting to the cancer research filed. Some minor concerns are listed below: Major points 1. In the experiments that the depletion of endogenous miR-205-5p affected the expression of BRCA1 and RAD17 proteins (Fig. 1G and H). It is interesting to note that the depletion of endogenous miR-205-5p affected the expression of BRCA1 more severely than the expression of RAD17 by luc reporter assay (Fig. 1G.) However, as assay by western blotting of BRCA1, RAD17 (Fig. 1H), It seems that the depletion of endogenous miR-205-5p affected the expression of RAD17 more severely than the expression of BRCA1. This contrast should clarify or discuss. 2. It is interesting to learn that whether other cancers, e.g., breast or ovary cancers, also with similar mechanisms. These should discuss in the discussion section. Minor points: 1. Line :……600.000 cases annually …… change to ……600,000 cases annually … 2. Line 118—Line 169: (Fig. 1a), change to (Fig. 1A); (Fig. 1b), change to (Fig. 1B)….. (Fig. 1i), change to (Fig. 1 I). And Fig. 2, Fig. 3, Fig. 4, Fig. 5 and Fig. 6 should modify as mentioned.

Author Response

Reviewer 2

Base on the authors previous findings (Elife. 2015; doi: 10.7554/eLife.05005; Cancer Res. 2007;67:2396-401. 727), this manuscript confirmed and documented that miR-205-5p, a miR highly expressed in Head and Neck Squamous Cell Carcinoma (HNSCC), targeted BRCA1 and RAD17 expression in vitro as well as in vivo. The results are fully support the mechanisms the authors try to pursuit and will be interesting to the cancer research filed. Some minor concerns are listed below:

Major points

In the experiments that the depletion of endogenous miR-205-5p affected the expression of BRCA1 and RAD17 proteins (Fig. 1G and H). It is interesting to note that the depletion of endogenous miR-205-5p affected the expression of BRCA1 more severely than the expression of RAD17 by luc reporter assay (Fig. 1G.) However, as assay by western blotting of BRCA1, RAD17 (Fig. 1H), It seems that the depletion of endogenous miR-205-5p affected the expression of RAD17 more severely than the expression of BRCA1. This contrast should clarify or discuss.

Response: We wish to thank the reviewer for this observation. The main of the luc reported assay was that of showing whether miR-205-5p targeting of RAD17 and BRCA1 was mediated by its specific seed on the respective 3'UTRs. Since, as described in Supplementary Materials and Methods section, the LUC-reporter vector for 3'-UTR-RAD17 (we cloned 702 bp from the RAD17-3'UTR region in the XhoI and NotI sites of psi-CHEK2TM - Promega) and 3'-UTR-BRCA1 (pMirTarget-3'UTR-BRCA1 was purchased by OriGene Technologies) are different, we avoided to make any assumption considering different affinity of miR205-5p to RAD17 and BRCA1 targets. We cannot exclude this possibility and consequently we have commented it in the discussion section of the revised manuscript.

It is interesting to learn that whether other cancers, e.g., breast or ovary cancers, also with similar mechanisms. These should discuss in the discussion section.

Response: We wish to thank the reviewer for this comment. We have preliminary findings showing that mutantp53 induces the transcription of the MIR205HG locus with the production of LNCRNA and miR-205-5p in the triple negative subgroup of breast cancers, as we have previously documented for head and neck cancers (Theranostics, Di Agostino et al., 2018). Aberrant expression of miR-205-5p is higher in serous ovarian cancers and associates with poor patient prognosis. Both cancer types are characterized by high genomic instability.
As requested by the reviewer a specific sentence has been added in the
discussion section.

Minor points: 1. Line :……600.000 cases annually …… change to ……600,000 cases annually … 2. Line 118—Line 169: (Fig. 1a), change to (Fig. 1A); (Fig. 1b), change to (Fig. 1B)….. (Fig. 1i), change to (Fig. 1 I). And Fig. 2, Fig. 3, Fig. 4, Fig. 5 and Fig. 6 should modify as mentioned.

Response: As requested by the reviewer we addressed these points in the revised version of the manuscript.